# Mass Spectrometry Analysis of Shark Skin Proteins

**DOI:** 10.3390/ijms242316954

**Published:** 2023-11-29

**Authors:** Etty Bachar-Wikstrom, Braham Dhillon, Navi Gill Dhillon, Lisa Abbo, Sara K. Lindén, Jakob D. Wikstrom

**Affiliations:** 1Dermatology and Venereology Division, Department of Medicine (Solna), Karolinska Institutet, 17177 Stockholm, Sweden; 2Whitman Center, Marine Biological Laboratory, Woods Hole, MA 02543, USA; 3Department of Plant Pathology, Fort Lauderdale Research and Education Center, IFAS, University of Florida, Davie, FL 33314, USA; 4Department of Biological Sciences, Nova Southeastern University, Davie, FL 33314, USA; 5Department of Medical Biochemistry and Cell Biology, Institute of Biomedicine, Sahlgrenska Academy, University of Gothenburg, 40530 Gothenburg, Sweden; 6Dermato-Venereology Clinic, Karolinska University Hospital, 17176 Stockholm, Sweden

**Keywords:** elasmobranchs, sharks, skin, mucin, proteins, mass spectrometry, proteomics, mucus layer

## Abstract

The mucus layer covering the skin of fish has several roles, including protection against pathogens and mechanical damage in which proteins play a key role. While proteins in the skin mucus layer of various common bony fish species have been explored, the proteins of shark skin mucus remain unexplored. In this pilot study, we examine the protein composition of the skin mucus in spiny dogfish sharks and chain catsharks through mass spectrometry (NanoLC-MS/MS). Overall, we identified 206 and 72 proteins in spiny dogfish (*Squalus acanthias)* and chain catsharks (*Scyliorhinus retifer*), respectively. Categorization showed that the proteins belonged to diverse biological processes and that most proteins were cellular albeit a significant minority were secreted, indicative of mucosal immune roles. The secreted proteins are reviewed in detail with emphasis on their immune potentials. Moreover, STRING protein–protein association network analysis showed that proteins of closely related shark species were more similar as compared to a more distantly related shark and a bony fish, although there were also significant overlaps. This study contributes to the growing field of molecular shark studies and provides a foundation for further research into the functional roles and potential human biomedical implications of shark skin mucus proteins.

## 1. Introduction

Elasmobranchs, encompassing sharks as a prominent example, have garnered considerable research focus owing to conservation endeavors. However, their molecular biology remains a subject of immense scientific interest, despite the inherent challenges associated with experimental investigations. Prior investigations have yielded noteworthy findings with potential implications for human medicine. Notably, the liver and stomach of Atlantic spiny dogfish (*Squalus acanthias*) sharks unveiled the presence of the antibiotic squalamine [1], while research on chloride channels within the rectal gland of these sharks [2] has proven relevant to the study of cystic fibrosis. Furthermore, the denticle patterns found on the skin of shortfin mako sharks (*Isurus oxyrinchus*) have proven to be valuable in enhancing the aerodynamic properties of airplanes, specifically in terms of reducing drag and improving lift, which bears resemblance to how these denticle patterns enhance the swimming abilities of sharks in water [3].

One significant contrast between fish and mammalian skin is that the dead, keratinized protective layer of skin known as the stratum corneum is absent in nearly all fish species. Instead, fish epidermis is composed solely of living cells [4,5] and is shielded by a layer of mucus. This slimy substance comprises large glycoproteins called mucins that creates a scaffold for multiple other proteins and glycoproteins, some of which exhibit antimicrobial properties that aid in preventing the entry and establishment of pathogens [4,6,7]. In mammalian infection models, increased susceptibility to infections have been demonstrated in animals lacking specific mucins [8,9,10,11]. The mucus layer is created both through secretion by various secretory cell types present in the epidermis as well as sloughing of dead cells [12].

Despite the well-documented proteome of the mucus layer in certain common bony fish (Osteichthyes), likely due to the demands of the fish farming industry, our understanding of the proteome within elasmobranchs, the main subclass of cartilaginous fish (Chondrichthyes) including most sharks, remains limited. Shark skin boasts unique attributes, including its tooth-like denticles, suggesting the possibility of novel proteins in the mucus layer with distinctive properties and functions, such as pathogen defense. A characterization of the proteome in shark mucus represents a crucial initial stride toward unraveling its biological significance. Using liquid chromatography–electrospray ionization tandem mass spectrometry, we herein present the most comprehensive proteome description of shark mucus to date in two shark species representing different genera: Atlantic spiny dogfish (*Squalus acanthias*), one of the most common shark species worldwide, and the bottom-dwelling chain catshark (*Scyliorhinus retifer*).

## 2. Results and Discussion

Over the preceding decade, there has been a notable increase in the number of studies focused on Chondrichthyes, encompassing sharks and rays. These investigations have sought to elucidate their biological intricacies, primarily with a focus on conservation-oriented research [13,14,15]. Nevertheless, despite the significant importance of exploring Chondrichthyes for conservation and preservation efforts, as well as for translating their exceptional features for potential therapeutic applications, molecular understanding of sharks remains comparatively limited when juxtaposed with bony fish and mammals.

The skin mucus of the more prevalent bony fish (Osteichthyes) and its critical role in fish health has been examined utilizing proteomics methodologies in several studies. The primary emphasis has been on the identification and characterization of innate and adaptive immune system proteins [12,16,17,18,19]. Sharks differ from bony fish in several aspects including cartilaginous skeleton and skin structure characterized by placoid scales (denticles) that reduce fluid friction, thereby enhancing swimming efficiency [20]. The mucus layer of sharks is far less researched than in bony fish and is probably different due to dissimilar skin architecture as well as extensive evolutionary separation. In sharks, proteomics studies are scarce [21,22,23,24] perhaps due to the fact that less than 1% of Chondrichthyan species have a sequenced genome [25], which complicates analysis combined with practical experimental difficulties in handling sharks. In this work, we present, for the first time, the identification of proteins from shark skin mucus using mass spectrometry, with a focus on immune-relevant molecules.

### 2.1. Protein Categorization

Skin protein samples were harvested from spiny dogfish and chain catsharks, as shown in Figure 1 and described in Methods. By utilizing nanoscale liquid chromatography coupled with tandem mass spectrometry (NanoLC-MS/MS) a wide range of proteins were identified from the skin mucus of both shark species. The protein fragments were matched against the Swissprot human database and Uniprot Chondrichthyes database using Mascot 2.5.1, a tool useful for proteome analyses of relatively unexplored species such as sharks, which have very little molecular data available in public databases. Overall, we identified 206 and 72 proteins in spiny dogfish and chain catsharks, respectively, described in detail in Table 1 and Table 2 as well as Appendix A. These proteins could, in principle, be from either cellular sloughing, a central process in skin physiology, or actively secreted to the mucus. The fact that more proteins were identified in spiny dogfish than catsharks is consistent with a previous study of ours in which less glycans were found in chain catshark skin and may be attributed to the tissue absorption sampling method, which was developed for teleost fish, or represent true biological differences such as a thinner mucus protein layer in chain catsharks [26].

Due to the importance of the mucus as a defense barrier, we focused our attention on the proteins that may have immune system roles. Proteins were grouped into eight different clusters based on biological process annotation as described in Methods (Table 3 for dogfish and catsharks, respectively). Carbohydrate and protein metabolism represented almost 40% of the proteins detected, possibly reflective of active regulatory processes such as osmoregulation, respiration, nutrition, or locomotion, as well as defense against pathogens [27], while immune-related proteins represented almost 20%, conceivably reflecting mucus antimicrobial properties. These proportions resemble previously reported data in Atlantic cod [28]. We further classified the proteins based on their type and found that 84% and 72% of the proteins in dogfish and catsharks, respectively, are cytoplasmic (including organelles and nucleus residential proteins) and that 19% and 31% are secreted proteins (Table 3 under “cellular location”), perhaps not surprising due to the fact that skin constantly turns over by sloughing. The secreted proteins are particularly interesting, as they include diverse classes of molecules such as mucins, immunoglobulins, proteases, and other proteins that have well-established roles in the immune system [29]. In Table 4 and Table 5 (secreted proteins), we characterize in detail the secreted proteins found in the sharks’ skin mucus and below we discuss their potential roles, together with other immune-related proteins from other classified groups (see Table 1 and Table 2).

Mucins are large glycoproteins that cover epithelial cell surfaces and form gel-like structures, thereby able to protect against harmful molecules and microorganisms. We found mucin-5B and 5B-like (catshark, dogfish, respectively), mucin-2-like (dogfish), as well as von Willebrand factor domain (vWFD)-containing protein (dogfish, catshark), which all are large, secreted gel-forming mucins harboring a cysteine-rich domain that strengthens the mucus barrier [60]; however, the vWFD can also be found in other, non-glycosylated, proteins. Thus, despite the sharks not appearing “slimy” as bony fish, they are indeed covered by mucins albeit with a thinner layer [26]. A study from 2013 on gilthead sea bream (*Sparus aurata*) skin showed that mucin-2 and mucin-2-like are expressed at relatively low levels and that probiotics [61] and bacterial infection [62] increased mucin-5B expression. Thus, these proteins may serve antimicrobial purposes in the shark skin. In a bioinformatic report from 2016 in which mucin protein sequences in several species were predicted from genomic sequences, mucin-5, 2, and 6 were identified in elephant shark (*Callorhinchus milii*), although not verified experimentally [63]. To the best of our knowledge, the present study is the first time these three mucins (mucin-5B, mucin-2, and mucin-2-like) are shown experimentally in shark skin mucus.

Jawed fish, including Osteichthyes (bony fish) and Chondrichthyes, are the most primitive animals that can make antibodies, however of different subclasses than mammals. Previously, several immunoglobulins (Ig) including IgM, IgH, IgD (IgW orthologue), and IgL were identified in bony fish, whereas three heavy chain isotopes including IgM, IgW, and immunoglobulin novel antigen receptor (IgNAR) were reported in Chondrichthyes [64]. Due to its small size, shark IgNAR is often referred to as a nanobody and is the primary antibody of a shark’s adaptive immune system with a serum concentration of 0.1–1.0 mg/mL. Shark IgNAR may have developed from the IgW gene [65] and was previously identified in spiny dogfish serum [66]. In spiny dogfish skin, we identified only the secreted IgW heavy chain (Table 1). IgW is believed to be the primordial antibody rather than IgM [67] and was first reported in the spleen of sandbar sharks (*Carcharhinus plumbeus*) in the early 1990s [41], and later in serum and lymphoid tissues of other sharks [68]. Although discovered before in spiny dogfish serum, as well as well as in other shark species organs such as pancreas [69], herein we report for the first time that IgW is present in the shark skin mucin as well, where it may serve an antimicrobial role.

Furthermore, we discovered proteins and enzymes such as GDP-L-fucose synthase, fucolectin tachylectin-4 pentraxin-1 (FTP) domain-containing protein (fragment) and GDP-mannose 4,6-dehydratase, which are involved in glycosylation, specifically fucosylation. Fucosylation is a glycan sugar protein modification essential to biological processes such as host–microbiota communication, viral infection or immunity [70]. We have previously shown that fucosylated glycans are common on spiny dogfish, chain catshark, and little skate skin mucus proteins [26]. Moreover, other proteins that are commonly post-translationally modified by glycosylation including antithrombin, fibrinogen beta chain, transferrin and serotransferrin, hemoglobin subunit alpha, syndecan binding protein, and cystatin kininogen-type domain-containing protein were also identified in this study. Apart from their well-known role in hemostasis, these proteins have a role in the activation of the immune system [37,38]. For instance, Ræder et al. [71] first identified a transferrin-like molecule in Atlantic salmon (*Salmo salar*) mucus infected with *Vibrio salmonicida*. The primary role of transferrin, which is a glycoprotein, is to sequester iron in a redox-inactive form making iron unavailable to pathogens, thus starving them [72]. This is probably important in the sharks’ skin antimicrobial defense, as iron is very limited in sea water [73].

The complement system, a network of more than 50 plasma and membrane-associated proteins, plays a vital role in vertebrate defense against pathogens in the blood as part of the innate and adaptive immune system [74]. Upon activation, the intermediate key factor, complement component (C3), acts as a chemoattractant, phagocytotic agent and as agglutinin and initiates a cascade of events leading to bacterial lysis and also acts as an inflammation mediator. While most studies on the complement system have been carried out on blood and internal organs, it has been described to be active in the skin as well since human keratinocytes infected with intracellular *Staphylococcus aureus* can be attacked by the complement system [75]. Notably, in the skin, complement dysregulation, deficiency, and genetic polymorphisms have been associated with a number of diseases such as psoriasis and recurrent cutaneous infection [76]. In dogfish, we identified C3, as well as component 1Q (C1q) and complement protein 1S. In addition, sushi domain-containing protein, which binds complement factors, was found both in spiny dogfish and chain catsharks. The presence of these proteins points to an active immune system, in general, and active complement cascade, specifically, in the skin mucus of dogfish sharks. In fact, a report published as early as 1907 suggested the presence of complement-like activity in dogfish serum [77]. Sixty years later, Legler and Evans described the serum hemolytic complement activity in three elasmobranch species including sting ray (*Dasyatis americana*) and two species of shark, lemon shark (*Nagaparion brevirostria*) and nurse shark (*Ginglyraostoma cirraium*) [78]. The C1q protein (complement system member) was first reported in skin mucus of European sea bass (*Dicentrarchus labrax*) [19]. To our knowledge, our data are the first description of complement components being present in shark skin mucus where it may play an antibacterial role.

Lectins are proteins that bind to carbohydrates, for example, on bacteria, and serve multiple roles including antimicrobial and developmental [79]. Lectins have been reported from various tissues of many fish species including skin mucus [28]. We found four lectins in the shark skin mucus, including the following: (1) L-type lectin-containing protein (dogfish), which interacts with *N*-glycans (components of glycoproteins) in a Ca^2+^-dependent manner [80]. (2) Calreticulin (dogfish, catshark), which is important for the cell surface expression of MHC class I molecules and antigen recognition [81]. (3) F-type lectins such as fucolectin tachylectin-4 pentraxin-1 (FTP) domain-containing protein (catshark), which is implicated in innate immunity (Table 5, detailed explanation). Of note, F-type lectin has been discovered in several fish species in the liver, intestines, and eggs [82,83,84] but to date not in the skin and not in sharks; however, C-type lectin has been found in the skin of Japanese bullhead shark (*Heterodontus japonicus*) [85]. (4) Calmodulin (dogfish), which also is involved in immune and inflammatory responses [86].

**Table 5 ijms-24-16954-t005:** Secreted proteins identified in the mucus of chain catshark. A literature-based distinction of their immune potential. Organism represents the protein reference species.

Accession Number (UniProt)	Protein Name	Organism ^a^	Function
A0A401P1E2	Mucin-5B	CCS	Highly glycosylated and gel-forming macromolecular components of mucus secretions. [30]. Also named vWFD domain-containing protein, exhibiting an evolutionarily- conserved von Willebrand factor type D domain (vWD), found in mucins [31].
A0A401QGB0	FTP domain-containing protein (fragment)	CCS	Fucolectin tachylectin-4 pentraxin-1 domain-containing protein. Acts as a defensive agent and recognizes blood group fucosylated oligosaccharides including A, B, H, and Lewis B-type antigens (Uniprot) [87].
A0A401PKI6	GDP-mannose 4,6- dehydratase	CCS	See A0A4W3GT93
A0A401PH36	vWFD domain- containing protein	CCS	See A0A401PH36
A0A401NTT0	N(4)-(Beta-N-acetylglucosaminyl)-L- asparaginase	CCS	Has a role the catabolism of N-linked oligosaccharides of glycoproteins. It cleaves asparagine from N-acetylglucosamines as one of the final steps in the lysosomal breakdown of glycoproteins [88].
A0A401PMW9	N-acetylglucosamine-6-sulfatase	CCS	Degrades glycosaminoglycans such as heparin, heparan sulfate, and keratan sulfate [89].
A0A401NXV8	Intelectin	CCS	A Lectin that recognizes microbial carbohydrate chains in a Ca^+2^-dependent manner [90,91]. Binds to glycans from Gram-positive and Gram-negative bacteria, including *K.* *pneumoniae*, *S. pneumoniae*, *Y. pestis*, *P. mirabilis*, and *P.* *vulgaris* [91].
A0A401NHG1	Serotransferrin	CCS	See A0A401RZK0
A0A401QHD1	Annexin (fragment)	CCS	See A0A401SL10
A0A401PZD5	Annexin	CCS	See A0A401SL10
A0A401PS39	Annexin	CCS	See A0A401SL10
A0A401P412	Annexin (fragment)	CCS	See A0A401SL10
A0A401NHG1	Serotransferrin	CCS	See A0A401RZK0
H9LEQ0	Haptoglobin	NS	Acts as an antioxidant, has antibacterial activity, and plays a role in modulating the acute phase response (UniProt, [92]).
A0A401NHT1	Sushi domain-containing protein	CCS	See A0A401S7A0
A0A401NPB6	Cystatin kininogen-type domain-containing protein	CCS	Glycoproteins related to cystatins. This protein participates in blood coagulation, inflammatory response, and vasodilation [93].
A0A401PTT0	Cathepsin L (fragment)	CCS	Active enzyme in the extracellular space of antigen presenting cells (APCs) during inflammation [94].
A0A401P304	Argininosuccinate synthase	CCS	This enzyme channels extracellular L-arginine to nitric oxide synthesis pathway during inflammation [52].
Q6EE48	Cathepsin B (fragment)	SSCS	See A0A401PTT0
A0A401PKI6	GDP-mannose 4,6- dehydratase	CCS	See A0A4W3GT93
A0A401NTU8	Syndecan binding protein	CCS	A glycoprotein involved in the immune system activation [37,38].

^a^ CCS—cloudy catshark, *Scyliorhinus torazame*; NS—nurse shark, *Ginglymostoma cirratum* (*Squalus cirratus*); SSCS—small-spotted catshark, *Scyliorhinus canicula* (*Squalus canicula*).

Several proteasomes (protease complex, “genetic information processing” group, Table 1 and Table 2) were found in both shark types, whereas cysteine proteases such as cathepsin L and B protein were identified in catsharks (“protein metabolism” group, Table 2). Proteases are essential for activation of both the innate and adaptive immune systems and perform complement activation, initiation of proinflammatory responses and the generation of peptides from foreign antigens that are then presented to the major histocompatibility complex in the adaptive immune response [36,95]. Proteases have been detected in fish mucus of several cold water fish [96] as well as in fish preferring warmer waters such as the greater amberjack (*Seriola dumerili*), in which ectoparasite infection increases protease activity [97]. Proteases have also been found in the gut of bonnethead sharks (*Sphyrna tiburo*) where these contribute to food digestion [98]; however, these have not been studied in shark skin.

Several annexins were found in both sharks and are known to regulate the activities of innate immune cells, in particular the generation of proinflammatory mediators, as was described in Atlantic cod [99]. Although the role of annexins in the shark skin mucus has not been studied before, epigonal media derived from bonnethead sharks induced apoptosis in human cancer cells, possibly due to annexin as an apoptosis inducer [100], which highlights how sharks can be useful in human medicine.

Actin is one of the most prevalent proteins in eukaryote cells and has several roles including cell movement, cytoplasmic streaming, phagocytosis, and cytokinesis [101]. Several reports suggested that the presence of actin and other cytoskeleton related proteins may not simply be due to contamination from ruptured cells but may have a separate role in mucus structure and immune system [102,103,104]. In both shark types, we found cytoskeleton-related molecules (actin, filamin, tubulin, gelsolin, tropomyosin, septin, and keratin), which is not surprising, as these proteins are common and were probably sloughed off. However, these proteins may have immune-relevant function in shark mucus, as shown in Atlantic salmon for actin [103], in rainbow trout (*Salmo gairdneri*) for keratin [105], and in zebrafish for septin [106]. In addition, extracellular actin from insects can bind to bacteria and stimulate their killing by phagocytosis [104]. Thus, an intracellular protein may change role when extracellularly located on the skin surface. Moreover, cytoskeletal-related proteins identified in spiny dogfish seemed to participate in shark osmoregulatory tissues [22]. All together, these findings suggest that cytoskeletal proteins could be functionally active extracellularly in the shark skin mucus as well.

The 14-3-3 proteins are acidic proteins with several isoforms that are ubiquitously expressed, participate in regulatory processes, and are indirectly involved in immune response [28,107]. Ras-related proteins are involved in signal transduction, the regulation of several biological processes, and, aptly, the immune system [108]. In both shark types, we identified these proteins (“cell communication” group, Table 1 and Table 2). 14-3-3 was present in the skin mucus of several fish types [109], but its implication in fish skin (and shark skin) has yet to be determined. Ras proteins were shown to interact with parasite proteins in the skin mucus of common carp (*Ichthyophthirius multifiliis*) and thus might serve as a drug target [55]. As the mucus layer is formed both by secretion and cellular sloughing, proteomics will naturally identify both secretory, membranous as well as intracellular proteins. Of note is that proteins may have dual functions; for example, ribosomal proteins with antimicrobial properties have been identified in rainbow trout and cod skin [56,110].

### 2.2. Protein Interaction

Protein–protein interaction network analysis may shed a light on the predicted function of the identified proteins by revealing their interaction, as well as reveal how similar these interactions are to other species. For that purpose, we used the STRING database [111], and created a proteome interaction network by merging all the proteins identified from the skin mucus of the sharks investigated and compared to published orthologues from other shark and fish species (Figure 2, Figure 3, Figure 4 and Figure 5). The sources for the maps include interactions from the published literature describing experimentally studied interactions as well as databases. A confidence score for every protein–protein interaction was assigned to the network in which a higher score is assigned when an association is supported by several types of evidence. To minimize false positives as well as false negatives, all interactions tagged as “low confidence” (<0.4) in the STRING database were eliminated. Thus, the networks are composed of a set number of nodes (proteins) and edges (interactions) (Table 6 and Table 7). We found a much higher number of edges when comparing the spiny dogfish and chain catsharks to the phylogenetically close shark species cloudy catshark (*Scyliorhinus torazame*) and brownbanded bamboo shark (*Chiloscyllium punctatum*) as compared to the much more distant elephant shark (*Callorhinchus milii*) and zebrafish (*Danio rerio*) [112]. There is also a similar pattern in number of nodes (proteins) albeit less significant. From the STRING analysis, examining the percentage of proteins that had orthologues with other species revealed that (1) cloudy catshark overlaps 88% with dogfish and 99% with chain catsharks, which suggests that the two catshark species are closely related; (2) brownbanded bamboo shark overlaps 88% with dogfish and 93% with chain catsharks; (3) elephant shark overlaps 81% with dogfish and 89% with catshark, somewhat counterintuitive, as elephant sharks are phylogenetically distant; and (4) zebrafish overlaps 82% with dogfish and 86% with catsharks. Furthermore, up to 19% of the proteins did not have orthologues in other shark species, which could mean that they are unique in the respective sharks or, alternatively, this could be due to methodological differences. These data indicate that most of the skin mucus proteome is conserved and shared among close (although separated by millions of years of evolution) shark species and also a bony fish, and, while speculative, this argues that these proteins may serve important physiological functions. To determine whether the skin proteomes of different species evolved independently (convergent evolution) or were already present earlier in evolution, one would need to sample common ancestors such as coelacanths.

### 2.3. Therapeutic Implications and Human Relevance

Several studies have suggested that sharks may be relevant for human medicine. A recent comparison of gene transcripts between white shark (*Carcharodon carcharias*) and zebrafish revealed, surprisingly, that white shark gene products associated with metabolism, molecular functions, and the cellular locations of these functions were more similar to humans than to zebrafish [113]. Moreover, squalamine, a compound with a broad-spectrum antifungal, antibacterial, and antitumor activity, that was isolated from spiny dogfish tissues [1] has resulted in a phase I and phase II human trials [114,115]. Proteoglycans with anti-osteoarthritic properties isolated from the bramble shark (*Echinorhinus brucus*) cartilage showed significant improvement in disease parameters in an osteoarthritis rat model [116]. Elasmobranchs immunoglobulins and nanobodies (small monoclonal antibodies) have raised a great attention from the scientific community, as they are the earliest jawed vertebrates to possess all the components necessary to perform responses associated with the adaptive immune system [117]. The topic of sharks’ usefulness in human medicine was elegantly reviewed by Luer and Walsh [117].

### 2.4. Study Limitations

Only female spiny dogfish were sampled in this study. The mucus harvest method may have missed some proteins and did not work well in chain catsharks in which longer absorption time or scraping may be needed. Furthermore, the mass spectrometry analysis used only shows already known proteins; thus, novel proteins unique to sharks may have been missed, and complementary methods for novel protein discovery will thus need to be used in the future.

## 3. Methods

### 3.1. Animals

Spiny dogfish caught using hook gear were purchased from a commercial fisherman in Chatham, MA in 2022. Only female spiny dogfish were available, likely due to commercial fishing often targeting female schools [118]. Chain catsharks were collected from a National Oceanic and Atmospheric Administration survey vessel by dredging in the mid-north Atlantic between 2017 and 2019. All elasmobranchs were housed in tanks with natural sea water flow-through systems, maintained year-round at 14 °C at the Marine Resources Center (MRC) at the MBL. Elasmobranchs were housed in single-species groups and fed a diet of food-grade frozen capelin (Atlantic-Pacific North Kingstown, RI, USA) and fresh, frozen, locally caught squid three days per week. Photos were taken with an iPhone 13 Pro (Apple Inc., Cupertino, CA, USA)).

Experiments were approved by the Institutional Animal Care and Use Committee (IACUC) at the MBL (protocol no 22-22).

### 3.2. Skin Mucus Sampling

Skin mucus were sampled using the Kleenex tissue absorption method, previously developed for salmonoids [119]. Briefly, housed elasmobranchs were caught gently with a net and a Kleenex tissue was placed on the skin for 10 s whereafter the tissue was put in Spin-X tubes (Sigma-Aldrich, St. Louis, MO, USA) on ice and later spun down at 700 g in a 4 °C cooled benchtop centrifuge. Tank water controls samples was also harvested by placing the Kleenex (Kimberly-Clark, Irving, TX, USA) briefly in the tank water. The liquid samples were transferred to plastic cryotubes, snap-frozen on dry ice, and stored at −80 °C.

### 3.3. Proteomics

#### 3.3.1. Sample Preparation

Protein content was determined using a colorimetric assay (Bradford protein assay, Bio-Rad, Hercules, CA, USA). Aliquots corresponding to 20 µg protein were processed using a modified version of filter-aided sample preparation (FASP) method [120]. The mucus samples were reduced in reduction buffer (6 M GuHCl (guanidinum hydrochloride (ultrapure, MP Biomedicals, Santa Ana, CA, USA), 0.1 M TEAB (triethyl ammonium buffer pH 9.5), 5 mM ethylenediaminetetraacetic acid, 0.1 M dithiothreitol) for 30 min at 37 °C. The samples were transferred to 10 kDa Microcon Centrifugal Filter Units (MPE030025, polyetylensulfon filter, Millipore, Burlington, MA, USA), and washed repeatedly with 6M GuHCl, followed by alkylation with 100 µL 0.05 M iodoacetamide in 50 mM TEAB buffer for 30 min. Digestion was performed in 0.1M TEAB with the addition of sequencing-grade modified trypsin (Promega, Madison, WI, USA) in an enzyme-to-protein ratio of 1:100 at 37 °C overnight. An additional portion of trypsin was added and incubated for 4 h. Peptides were collected by centrifugation, followed by further purification using High Protein and Peptide Recovery Detergent Removal Spin Column and Pierce peptide desalting spin columns (both Thermo Fischer Scientific, Waltham, MA, USA) according to the manufacturer’s instructions.

#### 3.3.2. NanoLC/MS

NanoLC-MS/MS was performed on an Orbitrap Exploris 480 mass spectrometer interfaced with Easy-nLC1200 liquid chromatography system (both Thermo Fisher Scientific, Waltham, MA, USA). Peptides were trapped on an Acclaim Pepmap 100 C18 trap column (100 μm × 2 cm, particle size 5 μm, Thermo Fischer Scientific, Waltham, MA, USA) and separated on an in-house packed analytical column (75 μm × 35 cm, particle size 3 μm, Reprosil-Pur C18, Dr. Maisch) using a gradient from 5% to 35% ACN in 0.2% formic acid over 40 min at a flow of 300 nL/min. Each preparation was analyzed using MS1 scans settings, *m*/*z* 380–1500, at a resolution of 120 K. MS2 analysis was performed in a data-dependent mode at a resolution of 30K, using a cycle time of 2 s. The most abundant precursors with charges 2–6 were selected for fragmentation using HCD at collision energy settings of 30. The isolation window was set to 1.2 *m*/*z* and the dynamic exclusion was set to 10 ppm for 30 s.

#### 3.3.3. Proteomic Data Analysis

The acquired data were analyzed using Proteome Discoverer 2.4 (Thermo Fisher Scientific, Waltham, MA, USA). The raw files were matched against the Swissprot human database (March 2021) and Uniprot Chondrichthyes database (142,499 entries, February 2023) using Mascot 2.5.1 (Matrix Science, London, UK) as a database search engine with peptide tolerance of 5 ppm and fragment ion tolerance of 30 mmu. Tryptic peptides were accepted with one missed cleavage, mono-oxidation on methionine was set as a variable modification, and carbamidomethylation on cysteine was set as a fixed modification. Target Decoy was used for PSM validation. Tables referring to secreted proteins are based on targeted literature searches and UniProt data (www.uniprot.org (accessed on 1 June 2023)).

The proteins identified were clustered into different categories based on Gene Ontology category, biological process. Further classification of protein type and functional hierarchies of biological entities were based on information on KEGG BRITE Database (kegg.jp/kegg/brite.html (accessed on 1 June 2023)) and UniProt (uniprot.org (accessed on 1 June 2023)) for individual proteins. As most of the proteins are not well annotated in teleost species, the Gene Ontology terms were retrieved from its human counterparts.

### 3.4. Protein–Protein Interaction Network Analysis

Protein interaction network maps for the sharks’ skin mucus proteins was generated using STRING (https://version-12-0.string-db.org/ (accessed on 1 September 2023), employing the following organism UniProt IDs: Cloudy catshark (*Scyliorhinnus torazame*), Brownbanded bamboo shark (*Chiloscyllium punctatum*), Elephant shark (*Callorhinchus milii,* also called Australian ghost shark), Zebrafish (*Danio rerio*). To achieve a more stringent analysis, the active interaction sources were limited to experiments and databases, and an interaction score >0.4 was applied to construct the protein–protein interaction network.

### 3.5. Chemicals

The chemicals were from Sigma-Aldrich (St. Louis, MO, USA) unless stated otherwise.

## 4. Conclusions

This is the first study that describes the skin mucus proteome of sharks. These proteins represent several basic functional groups, and while most of them are cellular proteins, a substantial minority are secreted. We propose these skin proteomes to be relatively conserved between close shark species. Further research on elasmobranch skin is warranted, especially bioprospecting studies that aim to identify completely novel molecules using protein sequencing, decipher their functions experimentally, and, if possible, translate to human clinical use albeit with shark conservation in mind.

## Figures and Tables

**Figure 1 ijms-24-16954-f001:**
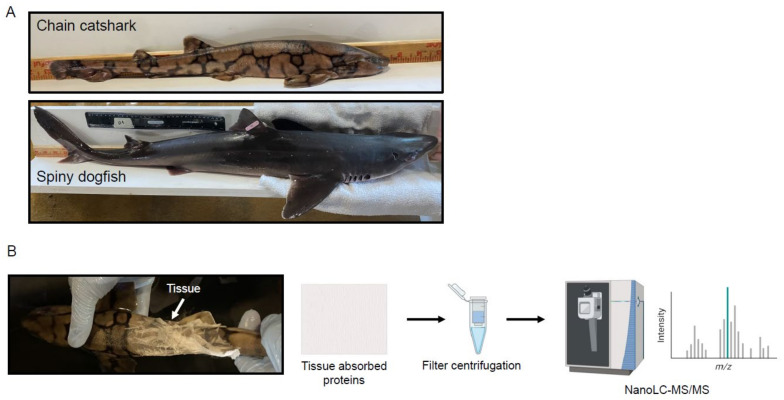
Experimental setup. (**A**) Shark species examined. The chain catshark scale is in inches and the spiny dogfish scale is in cm. (**B**) Sample harvest and analysis. Proteins were harvested by wrapping wet shark skin with a Kleenex tissue for 10 s, followed by centrifugation in SpinX tubes and analysis using mass spectrometry (NanoLC-MS/MS). N = 10 for spiny dogfish and N = 10 for chain catsharks.

**Figure 2 ijms-24-16954-f002:**
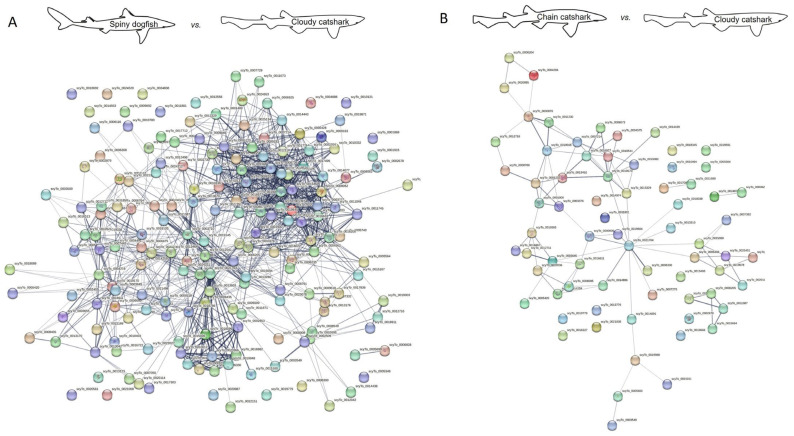
Protein interaction map of identified spiny dogfish (**A**) and chain catshark (**B**) skin proteins using cloudy catshark orthologues. A possible protein–protein interaction map with high edge confidence was generated using STRING. Ticker edges (line joining the nodes) represent a confidence of 0.4. Edges represent protein–protein association where association does not necessarily mean physical binding of the proteins and there could be involvement of several proteins to a shared function. Note that colored nodes represent different clusters of the query proteins, as employed by STRING software. Full protein names for the abbreviations are provided in Appendix A. Note that the larger number of proteins identified in dogfish relative to catsharks yields more interactions; for relative comparisons, see Table 6 and Table 7.

**Figure 3 ijms-24-16954-f003:**
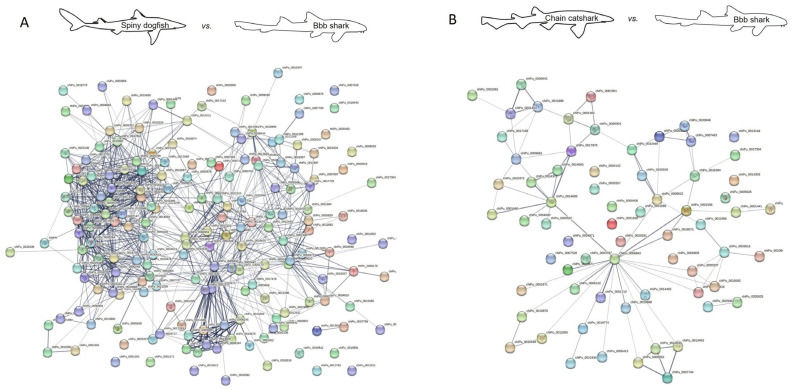
Protein interaction map of identified spiny dogfish (**A**) and chain catshark (**B**) skin proteins using brownbanded bamboo (bbb) shark orthologues. A possible protein–protein interaction map with high edge confidence was generated using STRING. Ticker edges (line joining the nodes) represent a confidence of 0.4. Edges represent protein–protein association where association does not necessarily mean physical binding of the proteins and there could be involvement of several proteins to a shared function. Note that colored nodes represent different clusters of the query proteins, as employed by STRING software. Full protein names for the abbreviations are provided in Appendix A. Note that the larger number of proteins identified in dogfish relative to catsharks yields more interactions; for relative comparisons, see Table 6 and Table 7.

**Figure 4 ijms-24-16954-f004:**
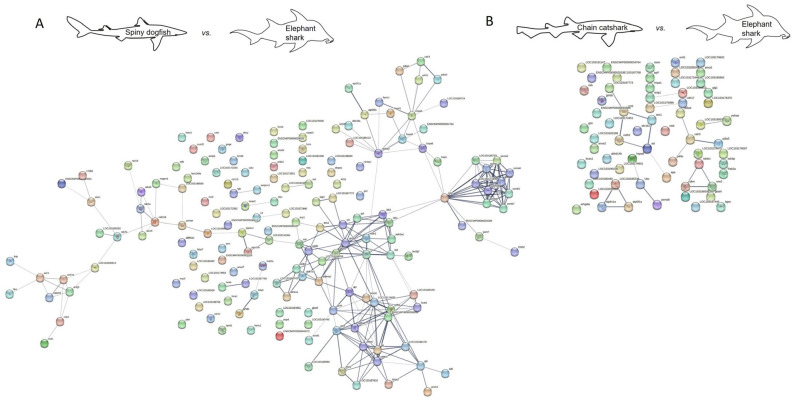
Protein interaction map of identified spiny dogfish (**A**) and chain catshark (**B**) skin proteins using elephant shark orthologues. A possible protein–protein interaction map with high edge confidence was generated using STRING. Ticker edges (line joining the nodes) represent a confidence of 0.4. Edges represent protein–protein association where association does not necessarily mean physical binding of the proteins and there could be involvement of several proteins to a shared function. Note that colored nodes represent different clusters of the query proteins, as employed by STRING software. Full protein names for the abbreviations are provided in Appendix A. Note that the larger number of proteins identified in dogfish relative to catsharks yields more interactions; for relative comparisons, see Table 6 and Table 7.

**Figure 5 ijms-24-16954-f005:**
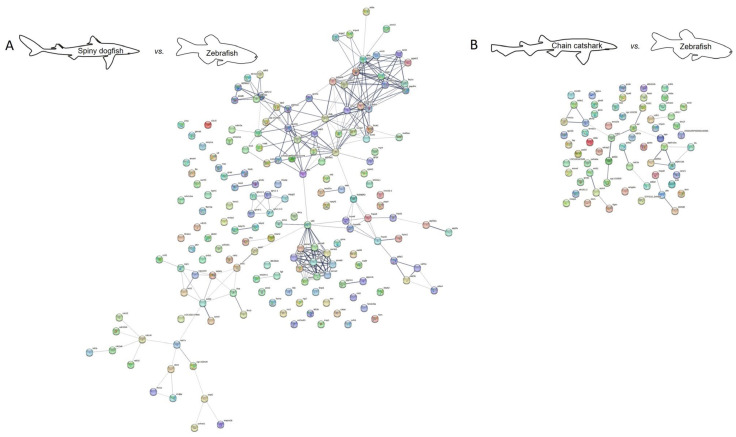
Protein interaction map of identified spiny dogfish (**A**) and chain catshark (**B**) skin proteins using zebrafish orthologues. A possible protein–protein interaction map with high edge confidence was generated using STRING. Ticker edges (line joining the nodes) represent a confidence of 0.4. Edges represent protein–protein association where association does not necessarily mean physical binding of the proteins and there could be involvement of several proteins to a shared function. Note that colored nodes represent different clusters of the query proteins, as employed by STRING software. Full protein names for the abbreviations are provided in Appendix A. Note that the larger number of proteins identified in dogfish relative to catsharks yields more interactions; for relative comparisons, see Table 5 and Table 6.

**Table 1 ijms-24-16954-t001:** Identified proteins from spiny dogfish skin mucus grouped into biological groups.

Accession Number	Immune-Related
A0A401T5Y9	Mucin-5B-like
A0A401RME8	Mucin-2-like
A0A401SE28	Ig-like domain-containing protein
A0A401PH36	vWFD domain-containing protein
P23085	Ig heavy chain C region (fragment)
A0A401NGS8	vWFA domain-containing protein
A0A0H4IU03	Antithrombin
A0A401RZK0	Serotransferrin
A0A401Q3Q5	GDP-L-fucose synthase (fragment)
A0A401RXA4	Prothymosin alpha
U5NJK8	Secreted IgW heavy chain
A0A401RMX1	Fibrinogen beta chain
A0A401RJ78	Complement component 1 Q subcomponent-binding protein, mitochondrial (fragment)
H9LDW9	Complement protein 1S
Q8HWH7	MHC class I antigen
P03983	Ig heavy chain V region
A0A401SX93	Prothymosin alpha-like
A0A088MN23	C3 complement component
A0A401P9G0	IRG1 decarboxylase (fragment)
A0A401Q3Q5	GDP-L-fucose synthase (fragment)
A0A401SNF0	Transferrin-like domain-containing protein
A0A401NHG1	Serotransferrin
A0A4W3I3U6	Serotransferrin
A0A401RZK4	Serotransferrin
**Accession Number**	**Genetic information processing**
A0A401SY06	Heat shock protein 70 (fragment)
A0A401S035	Proteasome subunit alpha type
K4G7F1	Proteasome subunit alpha type
A0A4W3H615	Proteasome subunit beta
A0A4W3I1Z3	Protein disulfide isomerase
A0A401NMQ4	Heat shock cognate 71 kDa protein
A0A401SNV8	Protein disulfide isomerase
A0A401RSV7	Proteasome subunit beta (fragment)
A0A401SEG3	60 kDa heat shock protein, mitochondrial
A0A401SGF6	Proteasome subunit alpha type
A0A401SL10	Annexin
A0A401RYU8	Proteasome subunit alpha type
V9KKG3	Annexin (fragment)
A0A4W3K8Z3	Protein disulfide isomerase
K4FY62	Proteasome subunit alpha type
A0A401PZD5	Annexin
A0A401SX58	Protein disulfide isomerase (fragment)
A0A401PL09	Proteasome subunit alpha type
A0A4W3GHI3	Proteasome 20S subunit alpha 2
Q9DEZ5	Ubiquitin (fragment)
A0A4W3IZD3	Tyrosine—tRNA ligase
A0A4W3IS83	GMP reductase
A0A4W3KHY8	RNA helicase
V9KVF9	Protein SET-like protein (fragment)
A0A401SFW5	Thioredoxin domain-containing protein
A0A4W3JL89	Calreticulin
A0A401STB5	Calreticulin
A0A4W3IN11	Thioredoxin disulfide reductase
A0A4W3HJ00	TNF receptor associated protein 1
A0A401NGY9	Calreticulin
V9LF43	Aminoacyl tRNA synthase complex-interacting multifunctional protein 1 (fragment)
A0A401SW93	Hypoxia upregulated 1
A0A401SF18	CN hydrolase domain-containing protein
A0A401PA35	78 kDa glucose-regulated protein
P27950	Nucleoside diphosphate kinase (fragment)
A0A401RY79	RING-type E3 ubiquitin transferase
A0A4W3JJD7	RAN binding protein 2
A0A4W3K2G6	Eukaryotic translation initiation factor 6
V9KJH1	Septin-2
A0A401QMS9	Quinolinate phosphoribosyltransferase (decarboxylating)
K4G9R8	Elongation factor 1-alpha
A0A401P0F5	Delta-1-pyrroline-5-carboxylate dehydrogenase, mitochondrial
A0A4W3JW47	Stress-70 protein, mitochondrial
A0A401PGN2	Myoferlin
A0A401NIS3	2-iminobutanoate/2-iminopropanoate deaminase
A0A4W3HQT2	Acyl-CoA dehydrogenase short chain
A0A401RYC2	Proliferating cell nuclear antigen
A0A4W3HJ00	TNF receptor associated protein 1
A0A401SVW0	Protein SET
A0A401NQG3	BPNT1 nucleotidase (fragment)
A0A401SK27	Calumenin
A0A411HEE0	Carbonic anhydrase
A0A401RS85	Phosphotriesterase-related protein
**Accession Number**	**Protein metabolism**
A0A401P304	Argininosuccinate synthase
A0A401NVA0	Protein arginine deiminase
K4G395	Serine/threonine protein phosphatase
A0A401RW52	Dipeptidyl peptidase 1
A0A401P906	Dipeptidyl peptidase 1
A0A401PKX8	Cytosol aminopeptidase
V9KUJ8	S-adenosylmethionine synthase
A0A4W3JRE6	Adenosylhomocysteinase
V9KVD7	Dimethylargininase (fragment)
A0A401S0W3	Serine/threonine protein phosphatase
A0A401SFC8	Protein arginine deiminase
A0A401QF75	Peptide-methionine (S)-S-oxide reductase (fragment)
A0A401Q2J1	Argininosuccinate lyase
A0A401SZD6	Aspartate aminotransferase
A0A401TGT7	alanine transaminase
A0A401SPB6	Glutamate dehydrogenase (NAD(P)(+))
A0A401S7A0	Sushi domain-containing protein
A0A401T2T9	Protein deglycase
A0A401NMQ6	Branched-chain amino acid aminotransferase
A0A401SIE1	Ornithine aminotransferase
A0A401Q121	Aspartyl aminopeptidase
A0A401SJG6	Calpastatin (fragment)
A0A401PT33	AMP_N domain-containing protein (fragment)
A0A4W3K041	Serine/threonine protein phosphatase
A0A401RNN1	Branched-chain amino acid aminotransferase
A0A401SFZ0	LRRcap domain-containing protein
A0A401SLZ2	2-oxoisovalerate dehydrogenase subunit alpha (fragment)
V9KNP8	N-acyl-aliphatic-L-amino acid amidohydrolase
A0A401NQW3	P/Homo B domain-containing protein
A0A401T7U8	FGE-sulfatase domain-containing protein (fragment)
A0A401S584	Inter-alpha-trypsin inhibitor heavy chain H3
A0A401RY12	CYTOSOL_AP domain-containing protein
A0A401T3E2	CYTOSOL_AP domain-containing protein
A0A401NQA1	GADL1 decarboxylase (fragment)
A0A401RXJ2	Prolyl endopeptidase
**Accession Number**	**Carbohydrate metabolism**
A0A401PAL6	Transaldolase
V9LII8	Inositol monophosphatase 2-like protein (fragment)
V9L3J0	Inositol-1-monophosphatase
A0A401QAH0	Neutral alpha-glucosidase AB (fragment)
A0A401PMW9	N-acetylglucosamine-6-sulfatase
A0A401NYQ0	6-phosphogluconate dehydrogenase, decarboxylating
A0A4W3JXK3	Glucose-6-phosphate isomerase
P00341	L-lactate dehydrogenase A chain
A0A4W3GT93	GDP-mannose 4,6-dehydratase
A0A401P2T7	Pyruvate dehydrogenase E1 component subunit beta
A0A4W3JPP5	Malate dehydrogenase, mitochondrial
A0A401P8K0	Pyr_redox_dim domain-containing protein
A0A401P258	S-(hydroxymethyl)glutathione dehydrogenase
A0A401NTL9	TRANSKETOLASE_1 domain-containing protein
A0A401Q0I1	Pyr_redox_2 domain-containing protein (fragment)
A0A401SRC5	Pyruvate dehydrogenase E1 component subunit alpha
A0A401PZL6	Malate dehydrogenase (fragment)
Q76BC4	Fructose-bisphosphate aldolase (fragment)
V9KVQ5	phosphopyruvate hydratase
A0A401SFI1	Succinate-CoA ligase (GDP-forming) subunit beta, mitochondrial
A0A401P9Y7	Inositol-1-monophosphatase
A0A401S408	Aldehyde dehydrogenase (NAD(+))
A0A401SF33	Succinate-semialdehyde dehydrogenase
A0A401PGK1	Pyruvate kinase
A0A401P3M5	Isocitrate dehydrogenase (NADP)
Q7ZZL2	Glyceraldehyde-3-phosphate dehydrogenase (fragment)
A0A401Q3Q5	GDP-L-fucose synthase (fragment)
A0A4W3GRJ7	Isocitrate dehydrogenase (NADP)
A0A401RXT3	Transaldolase
Q9DDG8	Phosphopyruvate hydratase (fragment)
A0A401PKI1	Beta-hexosaminidase
A0A401T325	Phosphomannomutase
A0A401NUA1	Fructose bisphosphatase
A0A4W3IQ20	Alcohol dehydrogenase (NADP(+))
A0A401PRM1	Aldedh domain-containing protein (fragment)
K4FRZ7	Aldehyde dehydrogenase (NAD(+))
A0A4W3JEX4	Aldehyde dehydrogenase 6 family member A1
A0A401SDE7	Glycine cleavage system H protein
A0A401RGC3	Citrate synthase
A0A401RJ17	Cis-aconitate decarboxylase
A0A401P0M9	L-type lectin-like domain-containing protein
V9KBN2	Transketolase
Q801K6	Triosephosphate isomerase (Fragment)
**Accession Number**	**Cell communication**
A0A4W3IRK1	14-3-3 protein zeta
K4FRV3	Ras-related protein ORAB-1
V9KM01	14-3-3 protein epsilon
A0A401RKJ5	Ras-related protein Rab-10
A0A401P7K5	Ras-related protein Rab-14
A0A4W3KI96	RAB7A, member RAS oncogene family
A0A4W3I0U6	RAB11B, member RAS oncogene family
A0A401RKJ5	Ras-related protein Rab-10
A0A401P7K5	Ras-related protein Rab-14
A0A4W3IRK1	14-3-3 protein zeta
V9KM01	14-3-3 protein epsilon
A0A4W3I6S4	RAB15, member RAS oncogene family
A0A4W3KI96	RAB7A, member RAS oncogene family
A0A4W3I0U6	RAB11B, member RAS oncogene family
A0A401T632	Calmodulin
K4FRV3	Ras-related protein ORAB-1
A0A4W3IZ94	Family with sequence similarity 149 member A
A0A401NXB1	Adenosine kinase
A0A401RV12	ATP synthase subunit beta
A0A401SNF0	Transferrin-like domain-containing protein
A0A401SSE0	Reticulocalbin-2
Q000H3	Superoxide dismutase (Cu-Zn)
A0A401RP32	LAMC1 protein (fragment)
A0A401NHG1	Serotransferrin
A0A401SSQ1	ATP synthase subunit alpha
A0A4W3I3U6	Serotransferrin
C0HJZ2	Hemoglobin subunit alpha (fragment)
A0A4W3JCS1	Histidine—tRNA ligase
A0A401T5K2	Histidine—tRNA ligase
A0A401RZK4	Serotransferrin
V9L0X4	RAN-binding protein 1
**Accession Number**	**Cytoskeleton-related**
A0A401PU26	Actin
A0A401SSQ7	Tropomyosin 1
A0A401NHE5	Tropomyosin 1
A0A401P520	Actinin alpha 4
A0A401S817	F-actin-capping protein subunit alpha
A0A401TJ26	Tropomyosin (fragment) O
K4G4H2	Tubulin beta chain
A0A401RP09	Cadherin-1
A0A401SRU2	PHB domain-containing protein
I0J0X5	Beta actin (fragment)
A0A401PTK4	MICOS complex subunit (fragment)
A0A401NQQ8	Adenylyl cyclase-associated protein
A0A401NRV5	Fascin
A0A401Q7Q0	Filamin-A (fragment)
A0A4W3J6W4	Filamin B
V9KC15	Fascin
A0A401PKY4	LIM and SH3 domain protein 1
A0A401SZY2	ADF-H domain-containing protein
A0A4W3H117	Attractin
A0A4W3IN76	Glyoxalase domain-containing 4
A0A401Q4Q2	IF rod domain-containing protein
A0A401P7L8	Gelsolin
A0A401PMT2	LIM zinc-binding domain-containing protein
A0A401Q9B7	IF rod domain-containing protein (fragment)
A0A401SCI8	EB1 C-terminal domain-containing protein
A0A401SX48	HP domain-containing protein
A0A401RYA2	Beta-centractin
**Accession Number**	**Lipid metabolism**
A0A401P0G6	FABP domain-containing protein
A0A401NMH8	FABP domain-containing protein
A0A4W3GDQ7	Copine-3-like
A0A401PJH8	Flotillin
**Accession Number**	**Others**
A0A401SQA2	Structural protein
A0A401RDW7	Uncharacterized protein
A0A401QEW8	Uncharacterized protein (fragment)

**Table 2 ijms-24-16954-t002:** Identified proteins from chain catshark skin mucus grouped into biological groups.

Accession Number	Immune-Related
A0A401P1E2	Mucin-5B
A0A401QGB0	FTP domain-containing protein (fragment)
A0A401PKI6	GDP-mannose 4,6-dehydratase
A0A401PH36	vWFD domain-containing protein
A0A401NTT0	N(4)-(Beta-N-acetylglucosaminyl)-L-asparaginase
A0A401PMW9	N-acetylglucosamine-6-sulfatase
A0A401NXV8	Intelectin
A0A401NHG1	Serotransferrin
A0A401NTT0	N(4)-(Beta-N-acetylglucosaminyl)-L-asparaginase
**Accession Number**	**Genetic information processing**
A0A401NGY9	Calreticulin
A0A401QHD1	Annexin (fragment)
A0A401PZD5	Annexin
A0A401PS39	Annexin
A0A401P412	Annexin (fragment)
A0A401SX58	Protein disulfide isomerase (fragment)
A0A401PWZ3	Protein disulfide isomerase
A0A401NT99	Protein disulfide isomerase (fragment)
A0A401PWD0	Peptidyl-prolyl cis-trans isomerase (fragment)
A0A401PGJ3	Protein disulfide isomerase
Q9DEZ5	Ubiquitin (fragment)
A0A401NX33	Zinc-binding protein A33-like
A0A401NHG1	Serotransferrin
A0A4W3HGD1	Heat shock cognate 71 kDa protein
A0A401RYU8	Proteasome subunit alpha type
**Accession Number**	**Protein metabolism**
H9LEQ0	Haptoglobin
A0A401S4Q4	Creatine kinase
K4GLE3	Dipeptidase B-like protein
A0A401NHT1	Sushi domain-containing protein
A0A401NPB6	Cystatin kininogen-type domain-containing protein
A0A401Q2J1	Argininosuccinate lyase
A0A401PTT0	Cathepsin L (fragment)
A0A401SZ08	Aspartate aminotransferase (fragment)
A0A401P304	Argininosuccinate synthase
Q6EE48	Cathepsin B (fragment)
A0A401NVA0	Protein arginine deiminase
A0A4W3J5I2	H(+)-transporting two-sector ATPase
A0A401SSQ1	ATP synthase subunit alpha
A0A401NZH2	Dipeptidyl peptidase IV membrane form (Fragment)
A0A401PMW9	N-acetylglucosamine-6-sulfatase
A0A401P4Q5	TGc domain-containing protein
A0A401RY28	Vacuolar proton pump subunit B
**Accession Number**	**Carbohydrate metabolism**
A0A401PMD4	Phosphopyruvate hydratase
A0A401NWX3	Malate dehydrogenase
A0A401PKI6	GDP-mannose 4,6-dehydratase
A0A401PGK1	Pyruvate kinase
A0A401P9Y7	Inositol-1-monophosphatase
A0A401NTT0	N(4)-(Beta-N-acetylglucosaminyl)-L-asparaginase
A0A401P1U4	Aldo_ket_red domain-containing protein
A0A401PPR5	Hyaluronidase (fragment)
A0A401PY00	AB hydrolase-1 domain-containing protein
A0A401PAL6	Transaldolase
**Accession Number**	**Cell communication**
V9KM01	14-3-3 protein epsilon
A0A401NTU8	Syndecan binding protein
A0A401SV03	Ras-related protein Rab-2A
A0A401PFE8	14_3_3 domain-containing protein
A0A401PHA8	Integrin_alpha2 domain-containing protein
A0A401PR00	C2 domain-containing protein (fragment)
A0A401PGG8	Rho GDP-dissociation inhibitor 1
**Accession Number**	**Cytoskeleton-related**
A0A401PN55	Golgi apparatus protein 1
A0A401NQV3	Cadherin-17
A0A401Q409	Cadherin-1
A0A401P520	Actinin alpha 4
A0A4W3HRB3	Keratin, type II cytoskeletal 8-like
A0A401Q538	Cadherin-1
A0A401NX75	HELP domain-containing protein
A0A401RHP4	Tropomyosin alpha-4 chain (fragment)
A0A401NHE5	Tropomyosin 1
A0A401P7L8	Gelsolin
A0A401NRJ6	Protein tyrosine phosphatase
A0A4W3IN76	Glyoxalase domain containing 4
A0A4W3JRG2	ACTB protein
A0A401PTK4	MICOS complex subunit (fragment)
A0A401SRU2	PHB domain-containing protein
A0A4W3JRG2	ACTB protein
**Accession Number**	**Lipid metabolism**
A0A401PDN1	Vitellogenin domain-containing protein
**Accession Number**	**Others**
A0A401S9B9	Breast carcinoma amplified sequence 1
A0A401Q2V9	UPAR/Ly6 domain-containing protein (fragment)
A0A401PTQ1	DUF3298 domain-containing protein (fragment)

**Table 3 ijms-24-16954-t003:** Classification of proteins from the shark skin mucus. Proteins from spiny dogfish and from chain catsharks identified using LC-MS/MS. The proteins were clustered into different categories based on the gene ontology category “biological process”. Further classification of protein type and cellular location) was carried out using UniProt data (www.oniprot.org) for individual proteins. Some proteins can be found in more than one cellular location and can also have more than one biological classification. Therefore, the sum of proteins from different classifications and locations can exceed the total number of proteins.

Spiny Dogfish	Classification	Number of Proteins	% of Total (206)
	Immune-related	24	11.6
	Genetic information processing	53	26
	Protein metabolism	35	17
	Carbohydrate metabolism	43	21
	Cell communication	31	15
	Cytoskeletal	27	13
	Lipid metabolism	4	2
	others	3	2
	**Cellular location**	**Number of proteins**	**% of total (206)**
	Secreted	39	18.9
	Cytoplasm (including organelles and nucleus)	173	84
	Membrane	25	12.1
**Chain catshark**	**Classification**	**Number of proteins**	**% of total (72)**
	Immune-related	9	13
	Genetic infomation processing	15	21
	Protein metabolism	17	21
	Carbohydrate metabolism	10	14
	Cell communication	7	10
	Cytoskeletal	16	22
	Lipid metabolism	1	1.4
	others	4	6
	**Cellular location**	**Number of proteins**	**% of total (72)**
	Secreted	22	30.6
	Cytoplasm (including organelles and nucleus)	52	72
	Membrane	17	23.6

**Table 4 ijms-24-16954-t004:** Secreted proteins identified in the mucus of spiny dogfish. A literature-based distinction of their immune potential. Organism represents the protein reference species.

Accession Number (UniProt)	Protein Name	Organism ^a^	Function
A0A401T5Y9	Mucin-5B-like	BBBS	Highly glycosylated and gel-forming macromolecular components of mucus secretions [30]. Also named vWFD domain-containing protein, exhibiting an evolutionarily-conserved von Willebrand factor type D domain (vWD), found in mucins [31].
A0A401RME8	Mucin-2-like	BBBS	Antimicrobial mucin gel that participates in innate immunity [32]. Also named vWFD domain-containing protein (see above).
A0A401SE28	Ig-like domain containing protein	BBBS	Immunoglobulin [33].
A0A401PH36	vWFD domain-containing protein	CCS	See A0A401T5Y9
P23085	Ig heavy chain C region (Fragment)	HS	Immunoglobulin (UniProt, [34]).
A0A401NGS8	vWFA domain-containing protein	CCS	Von Willebrand factor type A domain See A0A401T5Y9 and [34].
A0A0H4IU03	Antithrombin	BBBS	Regulates blood coagulation [35,36] and is involved in activation of the immune system [37,38]
A0A401RZK0	Serotransferrin	BBBS	Delivers iron to cells via a receptor-mediated endocytic process as well to remove toxic free iron from the blood and to provide an antibacterial, low-iron environment [39].
A0A401Q3Q5	GDP-L-fucose synthase (Fragment)	CCS	Involved in fucosylation [40].
U5NJK8	Secreted IgW heavy chain	NS	Immunoglobulin found in spiny dogfish serum [41].
A0A401RMX1	Fibrinogen beta chain	BBBS	β-component of fibrinogen, which serves key roles in hemostasis and antimicrobial host defense [42]
H9LDW9	Complement protein 1S	SDF	A component of the classical pathway of the complement system (UniProt, [43]).
P03983	Ig heavy chain V region	HS	V region of the variable domain of immunoglobulin heavy chains participates in the antigen recognition [44].
A0A088MN23	C3 complement component	NS	C3 plays a central role in the activation of the complement system [45]
A0A401Q3Q5	GDP-L-fucose synthase (Fragment)	CCS	See A0A401Q3Q5
A0A401SNF0	Transferrin-like domain-containing protein	BBBS	The transferrin-like domain contains conserved cysteine residues involved in disulfide bond formation [46].
A0A401NHG1	Serotransferrin	CCS	See A0A401RZK0
A0A4W3I3U6	Serotransferrin	GS	See A0A401RZK0
A0A401RZK4	Serotransferrin	BBBS	See A0A401RZK0
A0A401SL10	Annexin	BBBS	Plays important roles in the innate immune response as effector of glucocorticoid-mediated responses and regulator of the inflammatory process [47].
V9KKG3	Annexin (fragment)	GS	See A0A401SL10
A0A401PZD5	Annexin	CCS	See A0A401SL10
V9LF43	Aminoacyl tRNA synthase complex- interacting multifunctional protein 1 (fragment)	GS	A cytokine that is specifically induced by apoptosis, and it is involved in the control of angiogenesis, inflammation, and wound healing [48].
A0A401RS85	Phosphotriesterase- related protein	BBBS	Predicted to enable hydrolase activity, acting on ester bonds and zinc ion binding activity [49]
A0A401P906	Dipeptidyl peptidase 1	CCS	Lysosomal cysteine proteinase that activates serine proteinases in cells of the immune system [50].
V9KVD7	dimethylargininase (Fragment)	GS	Positive regulation of angiogenesis and vascular permeability (UniProt, [51].
A0A401Q2J1	Argininosuccinate lyase	CCS	Channels extracellular L-arginine to nitric oxide synthesis pathway during inflammation [52].
A0A401S7A0	Sushi domain-containing protein	BBBS	Sushi domains are known to be involved in many recognition processes, including the binding of several complement factors to fragments C3b and C4b [53]
A0A401S584	Inter-alpha-trypsin inhibitor heavy chain H3	BBBS	Heavy chain subunit of the pre-alpha-trypsin inhibitor complex. This complex stabilizes the extracellular matrix through its ability to bind hyaluronic acid, found in mucins (UniProt, [31]).
A0A4W3JXK3	Glucose-6-phosphate isomerase	GS	Induces immunoglobulin secretion [54]
A0A4W3GT93	GDP-mannose 4,6- dehydratase	GS	This enzyme converts GDP-mannose to GDP-4-dehydro-6-deoxy-D-mannose, the first of three steps for the conversion of GDP-mannose to GDP-fucose in animals, plants, and bacteria [55,56].
A0A401Q3Q5	GDP-L-fucose synthase (fragment)	CCS	See A0A401Q3Q5
Q9DDG8	Phosphopyruvate hydratase (fragment)	BBBS	Stimulates immunoglobulin production [57].
A0A401SNF0	Transferrin-like domain-containing protein	BBBS	See A0A401SNF0
A0A401RP32	LAMC1 protein (fragment)	BBSS	Role in cell adhesion, differentiation, migration, and signaling [58]
A0A401NHG1	Serotransferrin	CCS	See A0A401RZK0
A0A4W3I3U6	Serotransferrin	GS	See A0A401RZK0
C0HJZ2	Hemoglobin subunit alpha (fragment)	GLSS	Involved in oxygen transport from gills to the various peripheral tissues [59]
A0A401RZK4	Serotransferrin	BBBS	See A0A401RZK0

^a^ BBBS—brownbanded bamboo shark, *Chiloscyllium punctatum* (*Hemiscyllium punctatum*); GLSS—Greenland sleeper shark, *Somniosus microcephalus* (*Squalus microcephalus*); GS—ghost shark, *Callorhinchus milii*; CCS—cloudy catshark, *Scyliorhinus torazame*; HS—horn shark, *Heterodontus francisci* (*Cestracion francisci*); NS—nurse shark; *Ginglymostoma cirratum* (*Squalus cirratus*); SDF—spiny dogfish, *Squalus acanthias*.

**Table 6 ijms-24-16954-t006:** Spiny dogfish protein interaction summary table using STRING analysis. Proteins identified in skin mucus of spiny dogfish were analyzed when employing orthologues from four different species. Number of nodes depicts the number of orthologues found out of 206 proteins. Number of edges depicts the number of protein–protein interactions found with medium (0.4) confidence. Interaction source is shown both for all active sources (all) and also limited to experiments and databases for more stringent analysis.

	No. of Nodes	No. of Edges	Interaction Source
Spiny dogfish vs. zebrafish	170	831	All
	170	293	Experiments and databases
Spiny dogfish vs. cloudy catshark	183	1597	All
	183	1098	Experiments and databases
Spiny dogfish vs. brownbanded bamboo shark	182	1412	All
	182	959	Experiments and databases
Spiny dogfish vs. elephant shark	167	665	All
	167	271	Experiments and databases

**Table 7 ijms-24-16954-t007:** Chain catshark protein interaction summary table using STRING analysis. Proteins identified in skin mucus of chain catshark were analyzed when employing orthologues from four different species. Number of nodes depicts the number of orthologues found out of 72 proteins. Number of edges depicts the number of protein–protein interaction found with medium (0.4) confidence. Interaction source is shown both for all active sources (all) and also limited to experiments and databases for more stringent analysis.

	No. of Nodes	No. of Edges	Interaction Source
Chain catshark vs. zebrafish	62	62	All
	62	21	Experiments and databases
Chain catshark vs. cloudy catsharks	71	146	All
	71	94	Experiments and databases
Chain catshark vs. brownbanded bamboo shark	67	119	All
	67	82	Experiments and databases
Chain catshark vs. elephant shark	64	46	All
	64	20	Experiments and databases

## Data Availability

Some or all data from the study are available from the corresponding author by request.

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
