# Peer review of "Mass Spectrometry Analysis of Shark Skin Proteins"

_ijms, 2023, doi:10.3390/ijms242316954_

Round 1

Reviewer 1 Report

Comments and Suggestions for Authors

Overall, this is a marvelous manuscript that examines the protein content in mucus from two shark species. The topic is of interest as recent research on shark proteomics and metabolomics has revealed some potentially useful medicinal properties of compounds discovered in shark. More than 200 proteins are identified from the spiny dogfish and more than 70 in the chain catsharks. The manuscript is well-written and detailed, and the authors are especially commended for the scholarly Results and Discussion which provides a thorough analysis of their work and puts into context. Publication with minor revision is recommended, with one point to be addressed below.

1. Figure 2. There are some issues with this figure. First, Figure 2A and 2B are identical, and based on the reading of the text that should not be the case, as protein mapping in the spiny dogfish was more extensive (Table 1) than the chain catsharks (Table 2) so highly unlikely the distribution into classes is identical.

In addition, the pie charts do not add to 100%; this is likely because multiple proteins span more than one classification scheme. In these cases, I find pie charts to be ineffective and moreover confusing to a reader. The reviewer suggests instead of a Figure here, include a table that breaks down each species' proteins into biological processes as (x of 206) or (X of 72) for each species, and likewise for cellular location.

Overall, this is a strong manuscript that merits rapid publication.

Author Response

Overall, this is a marvelous manuscript that examines the protein content in mucus from two shark species. The topic is of interest as recent research on shark proteomics and metabolomics has revealed some potentially useful medicinal properties of compounds discovered in shark. More than 200 proteins are identified from the spiny dogfish and more than 70 in the chain catsharks. The manuscript is well-written and detailed, and the authors are especially commended for the scholarly Results and Discussion which provides a thorough analysis of their work and puts into context. Publication with minor revision is recommended, with one point to be addressed below.

Answer: We thank the reviewer for this positive outlook on the manuscript.

  1. Figure 2. There are some issues with this figure. First, Figure 2A and 2B are identical, and based on the reading of the text that should not be the case, as protein mapping in the spiny dogfish was more extensive (Table 1) than the chain catsharks (Table 2) so highly unlikely the distribution into classes is identical.

Answer:  Thanks for noticing this. There was a duplication mistake that is now corrected.

In addition, the pie charts do not add to 100%; this is likely because multiple proteins span more than one classification scheme. In these cases, I find pie charts to be ineffective and moreover confusing to a reader. The reviewer suggests instead of a Figure here, include a table that breaks down each species' proteins into biological processes as (x of 206) or (X of 72) for each species, and likewise for cellular location.

Answer: We have converted pie charts to tables and also explained in the legend of Table 3A-B why sums exceed 100%.

Overall, this is a strong manuscript that merits rapid publication.

Reviewer 2 Report

Comments and Suggestions for Authors

This is a pretty interesting and novel proteomics studies on several species of Chondrichthyes, which include manta rays and sharks.  The experiments appear to have been well executed, the data appears to be of high quality, and the subsequent analysis is of high quality and reasonably well explained throughout the manuscript. 

I do have a few questions about the methodology and the results, however:

1. Has the method of extraction shown to be comprehensive for fish skin proteins?  Is there an effect of the extraction on the efficiency of trypsin action

2. Is there a possible effect of glycan formation or glycosylation in general that can interfere with the detection of peptides derived from those skin proteins?  I would think that glycosylated peptides might elute quite early in the LC-MS/MS gradient and can even be lost by not being trapped by the trapping micro-column. 

3.  Related to this, glycosylation is a rare event in lysines and arginines, as far as I know. But can glycosylation in other sites interfere with trypsin-lysis?

4. Could the authors test the gene ontology analysis using annotated databases of other fish or more closely related animals?

5. It would be useful for biologists reading this paper that the authors explain how the annotation of the shark skin proteome was achieved, given the absence of published annotated proteomes.

6. The statement in lines 177-180 can be expanded further.  Is the explanation here that different sampling methods were used for the spiny dogfish than for the catshark samples?

7. Is the point of Figure 2 that the two not very related species of shark fish analyzed in this paper do indeed exhibit similar proteome distributions?

8. If the presence of fucosylated proteins was previously identified in a very recent paper, then I am not sure why the novel FTP domain protein and GDP mannose dehydratase were not identified in the previous study?

9. In lines 380-381, it is concluded that most of the skin proteome in sharks is both shared and conserved among close shark species.  If the species are separated by millions of years in origin, then this indicates that the skin proteome settled quite early during evolution, thus ruling out any form of convergent evolution, correct? 

10. Also, if the four fish species being compared are not closely-related species, then why there is such a high degree of overlap between the four proteomes (>80%)?

Author Response

First, we want to thank the reviewer for this extensive review. We have tried to answer the specific comments to the best of our abilities below.

1a. Has the method of extraction shown to be comprehensive for fish skin proteins? 

Answer:  Yes, this extraction method was previously used in salmons (PMID: 30923042) as well as several mammals (PMID: 30185674).

Is there an effect of the extraction on the efficiency of trypsin action?

1b.  No, trypsin action is not affected since the proteins are washed on the FASP filter.

  1. Is there a possible effect of glycan formation or glycosylation in general that can interfere with the detection of peptides derived from those skin proteins?  I would think that glycosylated peptides might elute quite early in the LC-MS/MS gradient and can even be lost by not being trapped by the trapping micro-column. 

Answer:  No, since the glycopeptides are not that hydrophilic. The retention in the separation is more affected by the amino acid sequence. To detect glycoproteins, other extraction and mass spectrometry protocols would need to be employed which is an interesting topic for future studies.

  1. Related to this, glycosylation is a rare event in lysines and arginines, as far as I know. But can glycosylation in other sites interfere with trypsin-lysis?

Answer:  Yes, but this potential issue is not relevant since glycosylated proteins are not detected with the extraction method and LC-MS/MS used in this study. Furthermore, digestion with trypsin alone is gold standard for LC-MS/MS. For future glycoproteomics one may use additional proteases such as chymotrypsin to cleave other amino acids.

  1. Could the authors test the gene ontology analysis using annotated databases of other fish or more closely related animals?

Answer:  The proteins identified were clustered into different categories based on gene ontology category, biological process. Further classification of protein type and functional hierarchies of biological entities were based on information on KEGG BRITE Database (kegg.jp/kegg/brite.html) and UniProt (uniprot.org) for individual proteins. As most of the proteins are not well annotated in teleost species the gene ontology terms were retrieved from its human counterparts. To our best knowledge there are no other databases available for sharks.

  1. It would be useful for biologists reading this paper that the authors explain how the annotation of the shark skin proteome was achieved, given the absence of published annotated proteomes.

Answer: Please see the answer to question 4.  Please note that annotation in less explored species like sharks is a challenge and because of this several databases were used.

  1. The statement in lines 177-180 can be expanded further.  Is the explanation here that different sampling methods were used for the spiny dogfish than for the catshark samples?

Answer: We have clarified this in line 181-183 (new text in track mode).

  1. Is the point of Figure 2 that the two not very related species of shark fish analyzed in this paper do indeed exhibit similar proteome distributions?

Answer: Thanks for noticing this duplication mistake. We have exchanged the faulty data with correct.

  1. If the presence of fucosylated proteins was previously identified in a very recent paper, then I am not sure why the novel FTP domain protein and GDP mannose dehydratase were not identified in the previous study?

Answer: The previous paper cited used methods for detecting glycans, not proteins per se such as the two mentioned in the question.

  1. In lines 380-381, it is concluded that most of the skin proteome in sharks is both shared and conserved among close shark species.  If the species are separated by millions of years in origin, then this indicates that the skin proteome settled quite early during evolution, thus ruling out any form of convergent evolution, correct? 

Answer: This is very good point. We have clarified this in line 390-392 (new text in track mode).

  1. Also, if the four fish species being compared are not closely-related species, then why there is such a high degree of overlap between the four proteomes (>80%)?

Answer: The fish skin proteome, which is part of the protective mucus layer, probably serves important functions including pathogen defense which are likely to be of general importance and thus may have appeared early in evolution, however this is speculative. Moreover, as stated in Study limitations: Furthermore, the mass spectrometry analysis used only shows already known proteins, thus novel proteins unique to sharks may have been missed, thus complementary methods for novel protein discovery will need to be used in the future.